# SuperMark: Robust and Training-free Image Watermarking via Diffusion-based Super-Resolution

## Abstract

In today's digital landscape, the intermingling of AI-generated and authentic content has heightened the importance of copyright protection and content authentication. Watermarking has emerged as a crucial technology to address these challenges, offering a general approach to safeguard both generated and real content. To be effective, watermarking methods must withstand various distortions and attacks. While current deep watermarking techniques typically employ an encoder–noise layer–decoder architecture and incorporate various distortions to enhance robustness, they often struggle to balance robustness and fidelity, and remain vulnerable to adaptive attacks, despite extensive training. To overcome these limitations, we propose SuperMark, a novel robust and training-free watermarking framework. Our approach draws inspiration from the parallels between watermark embedding/extraction in watermarking models and the denoising/noising processes in diffusion models. Specifically, SuperMark embeds the watermark into initial Gaussian noise using existing techniques and then applies pretrained Super-Resolution (SR) models to denoise the watermarked noise, producing the final watermarked image. For extraction, the process is reversed: the watermarked image is converted back to the initial watermarked noise via DDIM Inversion, from which the embedded watermark is then extracted. This flexible framework supports various noise injection methods and diffusion-based SR models, allowing for enhanced performance customization. The inherent robustness of the DDIM Inversion process against various perturbations enables SuperMark to demonstrate strong resilience to many distortions while maintaining high fidelity. Extensive experiments demonstrate SuperMark's effectiveness, achieving fidelity comparable to existing methods while significantly surpassing most in terms of robustness. Under normal distortions, SuperMark achieves an average watermark extraction bit accuracy of 99.46%, and 89.29% under adaptive attacks. Furthermore, SuperMark exhibits strong transferability across different datasets, SR models, watermark embedding methods, and resolutions.

## 1 Introduction

With the rapid advancement of text-to-image (T2I) models (Saharia et al., 2022; Rombach et al., 2022) and image-to-image (I2I) models (Brooks et al., 2023; Mokady et al., 2023), AI-generated content (AIGC) has been increasingly prevalent and harder to be distinguished from real images. To mitigate the challenges posed by this trend, various regulations (European Parliament, 2023; PBS NewsHour, 2024; Reuters, 2024) have emerged that mandate the embedding of watermarks into AI-generated images. These watermarks serve as a proactive measure for ensuring transparency, traceability, and copyright verification. There are two emerging approaches for watermarking: embedding watermarks during the image generation process (Fernandez et al., 2023; Wen et al., 2024; Yang et al., 2024) and applying watermarks to the generated images via post-processing (Rahman, 2013; Zhang et al., 2019; Jia et al., 2021). The latter approach is more flexible and general, as it can be applied to both AIGC and real images, which is the focus of this paper.

The performance of general watermarking is evaluated along two key dimensions: robustness and fidelity. Robustness refers to the watermark's ability to remain detectable and intact even when

Figure 1: (a) The pipeline of traditional watermarking methods, which are trained in an encoder-noise layer-decoder manner. (b) The pipeline of our proposed training-free SuperMark. Here, $m$ represents for the watermark information.

subjected to various distortions or attacks on the watermarked image, while fidelity means the visual consistency between the watermarked image and the cover image. Deep learning-based watermarking methods typically adopt an encoder-noise layer-decoder framework, introducing various distortions during training to enhance robustness. However, achieving a balance between strong robustness and high fidelity remains a significant challenge for these models. Furthermore, adaptive attacks (Zhao et al., 2023) based on VAE (Ballé et al., 2018; Cheng et al., 2020) and diffusion models (Brooks et al., 2023) can easily circumvent most existing watermarking methods. Although some works have attempted to enhance robustness against such attacks, they often require extensive training with carefully designed differentiable distortions (e.g., StegaStamp (Tancik et al., 2020), RoSteALS (Bui et al., 2023), and Robust-Wide (Hu et al., 2024)) or they compromise fidelity (e.g., StegaStamp (Tancik et al., 2020) and RoSteALS (Bui et al., 2023)).

We reveal that their limitations are mostly stemmed from the disentanglement between robustness and fidelity due to the the encoder–noise layer–decoder architecture and the joint training strategy. Moreover, we have two interesting observations: 1) there exists an inherent symmetry between the **embedding/extraction** of watermarks and the **denoising/noising** processes in diffusion models, and 2) diffusion process holds **inherent robustness** against different distortions. Based on these, we propose SuperMark to design a novel diffusion-based general watermarking framework, which can inherently achieve robustness and fidelity in a unified manner. Briefly, the embedding and extraction of a watermark essentially involve a reversible transformation between the watermark information and watermarked image. Similarly, in diffusion models, the processes of denoising and noising represent transformations between the Gaussian noise and sampled image. Leveraging this insight, SuperMark injects the watermark information into the initial Gaussian noise, and defaultly employs the Denoising Diffusion Implicit Model (DDIM) as the sampling method for watermark embedding, this process is deterministic and exhibits strong reversibility with the denoising process. Most importantly, its corresponding reversible process, known as DDIM Inversion, has demonstrated remarkable robustness against various perturbations (Wen et al., 2024; Yang et al., 2024). Thus, Super-Mark applies DDIM Inversion to cover the sampled image back into the initial watermarked noise for inherent robust extraction. To satisfy the fidelity requirement, we feed both the watermarked noise and the cover image into a pretrained diffusion-based Super-Resolution (SR) model to generate the watermarked image. Moreover, this entire process can be executed without any fine-tuning of the SR model. Figure 1 shows a comparison between SuperMark and the traditional watermarking framework.

Extensive experimental results demonstrate that SuperMark achieves strong robustness against both normal distortions (e.g., JPEG compression and Gaussian noise) and adaptive attacks (e.g., VAE-based and diffusion-based attacks). It achieves high watermark extraction accuracy, with 99.46% accuracy under normal distortions and 89.29% accuracy even under adaptive attacks on the MS-COCO dataset. Additionally, SuperMark maintains high fidelity, with a PSNR of 32.49 and an SSIM of 0.93. We also evaluate its transferability across different datasets, SR models, watermark injection methods, and image resolutions.

In summary, our key contributions are as follows:

- Our research uncovers a critical insight into current deep watermarking techniques: the encoder-noise layer-decoder architecture and joint training strategy create a trade-off between robustness and fidelity.

- We introduce SuperMark, a novel and training-free watermarking framework based on diffusion-based super-resolution models. SuperMark's simplicity and effectiveness allow it to seamlessly integrate with various watermark injection methods and pre-trained diffusion-based SR models.

- Extensive experiments demonstrate that SuperMark offers superior robustness against both normal distortions and adaptive attacks compared to most existing watermarking methods, while maintaining high fidelity.

## 2 BACKGROUND

### 2.1 DIFFUSION MODELS

Diffusion Models (DMs) are designed to predict and gradually remove varying levels of noise added to images during training. During inference, they iteratively denoise randomly sampled Gaussian noise $x_T \sim \mathcal{N}(0, 1)$, progressively generating high-quality images $x_0$. Denoising Diffusion Probabilistic Models (DDPMs (Ho et al., 2020)) are a widely-used implementation of DMs, but they typically require thousands of denoising steps to produce high-quality samples. To accelerate the sampling process, Denoising Diffusion Implicit Models (DDIMs (Song et al., 2021)) are proposed to improve DDPMs by introducing *a deterministic sampling process* that reduces the number of required steps while maintaining the quality of the generated data. Besides, DDIMs can encode from $x_0$ to $x_T$ and reconstruct $x_T$ from the resulting $x_0$ with low reconstruction error, a capability that DDPMs lack due to their stochastic nature. In other words, the transformation between $x_0$ and $x_T$ is reversible. The reverse process $x_T \rightarrow x_0$ is known as DDIM Inversion, which enables a wide range of applications, such as image editing (Mokady et al., 2023).

Despite these improvements in the sampling speed and efficiency, generating images directly in the pixel space remains computationally expensive in terms of both time and memory. To address it, Latent Diffusion Models (LDMs (Rombach et al., 2022)) are designed to operate in a compressed, lower-dimensional latent space, facilitated by the Variational Autoencoder (VAE) which could significantly reduce the costs. Super-Resolution (SR) models, an important application within Image-to-Image (I2I) tasks, can be also implemented using LDMs. The core idea is to concatenate a low-resolution image with a latent variable of the same resolution for denoising. The denoised latent variable is then decoded using a VAE decoder $\mathcal{D}$ to obtain the corresponding high-resolution image.

### 2.2 IMAGE SUPER-RESOLUTION WITH LATENT DIFFUSION

In this section, we provide a detailed explanation of how the latent diffusion-based super-resolution (SR) model $\mathcal{M}$ achieves image super-resolution. In general, $\mathcal{M}$ employs a Variational Autoencoder (VAE) to realize image resolution, using a scaling factor $f_{vae}$ defined as: $f_{vae} = \frac{S_I}{S_Z}$, where $S_I$ and $S_Z$ represent the size of the input image and its corresponding latent variable produced by the VAE encoder $\mathcal{E}$. The magnification factor $f_{sr}$ of $\mathcal{M}$ indicates the ratio by which the model increases the input image's resolution, which is equal to $f_{vae}$.

Specifically, given a low-resolution input image $I_{low}$ with dimensions $(C_{pixel}, H_{low}, W_{low})$, $\mathcal{M}$ performs iterative denoising as follows:

$$Z^0 = \text{Denoise}(\mathcal{M}(Z_{concat} = I_{low} \oplus Z^T)), \tag{1}$$

where $Z^0$ is the denoised latent variable with dimensions $(C_{latent}, H_{low}, W_{low})$, and $Z^T \sim \mathcal{N}(0, 1)$ is randomly sampled Gaussian noise of shape $(C_{latent}, H_{low}, W_{low})$. The input to the SR model, $Z_{concat}$, is formed by concatenating $I_{low}$ and $Z^T$, resulting in a shape of $(C_{pixel} + C_{latent}, H_{low}, W_{low})$. Here, $C_{pixel}$ refers to the number of pixel channels (typically 3 for RGB images), $C_{latent}$ is the number of latent channels in the VAE (e.g., 4 for SD-Upscaler), and $H_{low}$ and $W_{low}$ represent the height and width of $I_{low}$. The super-resolved image $I_{sr}$, with dimensions $(C_{pixel}, H_{high}, W_{high})$, is then produced by the VAE decoder $\mathcal{D}$: $I_{sr} = \mathcal{D}(Z^0)$. Since $H_{high} = f_{vae} \times H_{low}$, the magnification factor is given by $f_{sr} = \frac{H_{high}}{H_{low}} = f_{vae}$.

## 2.3 IMAGE WATERMARKING

Current image watermarking methods can mainly be divided into two categories: *in-generation* watermarking and *post-processing* watermarking. In-generation watermarking involves embedding watermarks during the image generation process of a target generative model and has emerged as a key approach alongside the rise of AI-generated content (AIGC). Two notable techniques in this field are Tree-Ring (Wen et al., 2024) and Gaussian Shading (Yang et al., 2024), both designed for diffusion-based text-to-image (T2I) models. These methods embed watermarks into the initial Gaussian noise and utilize inverted noise, obtained through DDIM Inversion, for watermark extraction. Specifically, Tree-Ring embeds multiple rings in the frequency domain center of the Gaussian noise and extracts the watermark from the same positions in the inverted noise's frequency domain. In contrast, Gaussian Shading samples Gaussian noise based on the watermark bit string and extracts the watermark by inverse sampling of the inverted noise. Both techniques have demonstrated strong robustness.

However, the aforementioned in-generation watermarking methods are limited to AI-generated images and are not the focus of this paper. Instead, we focus on post-processing watermarking methods, which can be applied to both real and generated images. Traditional robust post-processing methods, such as DwtDct (Rahman, 2013) and DwtDctSvd (Rahman, 2013), embed watermark messages into transformed domains, offering only limited robustness. With the rise of deep learning, new post-processing watermarking methods based on deep models have emerged to improve robustness. Most follow the encoder-noise layer-decoder framework, where the encoder embeds watermarks, and the decoder extracts them in the pixel space. Different methods use customized noise layers for specific robustness. For example, MBRS (Jia et al., 2021) enhances robustness against JPEG compression, while StegaStamp (Tancik et al., 2020) and PIMoG (Fang & et al., 2022) target robustness against physical distortions. SepMark (Wu et al., 2023) focuses on inpainting, and Robust-Wide (Hu et al., 2024) addresses instruction-driven image editing. Recently, RoSteALS (Bui et al., 2023) showed that embedding watermarks in the latent space of a VAE significantly boosts semantic robustness. Beyond the encoder-noise layer-decoder framework, ZoDiac (Zhang et al., 2024), similar to our approach, embeds watermarks in Gaussian noise for general robustness. However, ZoDiac requires optimizing the initial Gaussian noise for each image to ensure the denoised result closely matches the original. It then adds ring-shaped watermarks using the Tree-Ring method before generating the watermarked image with an unconditional diffusion model. This process adds optimization overhead and results in lower fidelity, distinguishing it from our approach.

## 3 METHODOLOGY

### 3.1 DESIGN PRINCIPLES

As illustrated in Figure 1, we compare the design principles of our proposed SuperMark with traditional watermarking methods. Traditional watermarking models typically rely on an encoder and decoder, both of which require extensive training to embed and extract watermarks. In contrast, the core component of our framework is a pre-trained diffusion model, which performs these tasks without additional fine-tuning. Furthermore, while traditional methods require an extra noise layer during training to enhance robustness, our approach leverages the inherent robustness of the diffusion process itself. Below, we discuss the considerations for selecting this diffusion model and how our framework effectively achieves watermark embedding and extraction.

**Watermark embedding stage:** In the traditional pipeline, the encoder takes the original image along with the watermark information and generates a watermarked image that closely resembles the original. For SuperMark, the diffusion model must be image-conditioned so that the denoised output closely matches the conditioned image. Our research indicates that diffusion-based super-resolution (SR) models, which generate higher-resolution versions of conditioned images, effectively meet this requirement. The Gaussian noise added to the conditioned image corresponds to the watermark information, allowing the watermark to be injected seamlessly. Techniques such as Gaussian Shading (Yang et al., 2024) and Tree-Ring (Wen et al., 2024) have already been developed to achieve this.

**Watermark extraction stage:** For traditional watermarking methods, the decoder is trained jointly with the encoder to extract the embedded watermark from the watermarked image. In contrast, SuperMark achieves this using the same SR model employed during watermark embedding. The

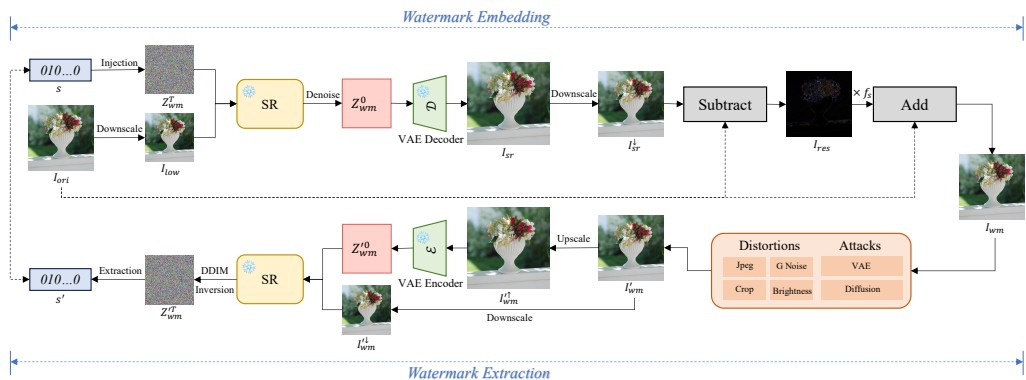

Figure 2: The end-to-end inference pipeline of SuperMark.

watermarked image is fed into the model, which performs DDIM Inversion to reconstruct the initial watermarked noise. From this reconstructed noise, the watermark can be extracted effectively, without requiring any additional training.

The flexibility of watermark injection into Gaussian noise and the choice of SR models are key strengths of SuperMark. These components can be interchanged and optimized, presenting exciting opportunities for future research and development.

### 3.2 OVERVIEW

The complete inference pipeline of SuperMark is illustrated in Figure 2, comprising two stages: watermark embedding and watermark extraction. Both the SR model $\mathcal{M}$ and the VAE operate with **frozen** parameters, meaning no additional training is required. In the watermark embedding stage, various techniques can be used to inject the watermark into the latent Gaussian noise, resulting in the watermarked noise $Z_{wm}^T$. This noise is then denoised to produce the watermarked image $I_{wm}$. In the watermark extraction stage, the distorted watermarked image $I_{wm}^{'}$ is processed using DDIM inversion to reconstruct the initial watermarked noise $Z_{wm}^{'T}$. From this reconstructed noise, the embedded watermark can be extracted. Below, we first present some preliminaries, followed by a detailed description of our method.

### 3.3 WATERMARK EMBEDDING

We adopt an off-the-shelf strategy for watermark embedding, as used in *Gaussian Shading* (Yang et al., 2024), with details provided in Appendix A.1. In general, the primary challenge of the watermark embedding process is to address the size discrepancy between the original image and the super-resolved image, while also balancing watermark robustness and image fidelity.

Due to the change in the size of the original image $I_{ori}$, caused by the SR model $\mathcal{M}$, the super-resolved image $I_{sr}$ cannot be directly used as the watermarked image. A straightforward solution would be to downscale $I_{sr}$ through interpolation to match the size of $I_{ori}$, using it as the watermarked image $I_{wm}$. However, this resizing process (e.g., downscaling by a factor of 1/4) results in the loss of a significant portion of watermarked pixels, greatly diminishing the robustness of the watermark. To address this issue, we downscale $I_{ori}$ to a smaller size before passing it through $\mathcal{M}$ for upscaling. This reduces the size discrepancy between $I_{sr}$ and $I_{ori}$, minimizing the loss of watermarked pixels during resizing and enhancing watermark robustness. However, downscaling $I_{ori}$ before inputting it into the model results in the loss of some original image details, which are then regenerated by the SR model. This leads to a trade-off between the robustness and fidelity, which we will explore in detail in Sec. 4.4.

The watermark embedding process is depicted in the upper half of Figure 2. We initially down resize $I_{ori}$, with a resolution of $H_{ori} \times W_{ori}$, to the image $I_{low}$ with a low resolution of $H_{low} \times W_{low}$. Subsequently, the watermark message $s$ can be injected into the Gaussian noise in various ways to obtain the watermarked Gaussian noise $Z_{wm}^T$. Then $I_{low}$ and $Z_{wm}^T$ are concatenated to a tensor as

$\mathcal{M}$'s input for iterative denoising to obtain the denoised watermarked latent $Z_{wm}^0$ which is converted to the super-resolved image $I_{sr}$ by the VAE decoder $\mathcal{D}$: $I_{sr} = \mathcal{D}(Z_{wm}^0)$. Afterwards, $I_{sr}$ with the resolution of $H_{high} \times W_{high}$ is down resized to acquire $I_{sr}^{\downarrow}$ with the resolution of $H_{ori} \times W_{ori}$. Now $I_{sr}^{\downarrow}$ and $I_{ori}$ have the same size, and we subtract them to get the residual image $I_{res}$:

$$I_{res} = I_{sr}^{\downarrow} - I_{ori}, \tag{2}$$

Finally, the watermarked image $I_{wm}$ is acquired by:

$$I_{wm} = I_{ori} + f_s \times I_{res}, \tag{3}$$

where $f_s$ is the strength factor used to balance the fidelity and robustness.

### 3.4 WATERMARK EXTRACTION

The watermark extraction process is illustrated in the lower part of Figure 2. In this process, $I_{wm}^{'}$ represents a distorted or attacked version of the original watermarked image $I_{wm}$. We explain how the watermark is extracted from $I_{wm}^{'}$ using DDIM Inversion below.

To perform DDIM Inversion using the model $\mathcal{M}$, the resolution of $I_{wm}^{'}$ must match the resolution used during the model's inference phase. To achieve this, we first upscale $I_{wm}^{'}$ to $I_{wm}^{'\uparrow}$, ensuring it matches the resolution of the super-resolved image, $I_{sr}$. The upscaled image is then encoded into the latent space using the VAE encoder $\mathcal{E}$, resulting in the latent representation $Z_{wm}^{'0}$: $Z_{wm}^{'0} = \mathcal{E}(I_{wm}^{'\uparrow})$. Additionally, we downscale $I_{wm}^{'}$ to $I_{wm}^{'\downarrow}$, matching the resolution of the original low-resolution image, $I_{low}$. Next, the super-resolution model $\mathcal{M}$ takes the concatenated tensor of the latent representation $Z_{wm}^{'0}$ and the downscaled image $I_{wm}^{'\downarrow}$ as input. It then performs DDIM Inversion to generate the reverted latent representation of the watermarked image, $Z_{wm}^{'T}$. Finally, depending on the specific watermark injection method used, the watermark is extracted from $Z_{wm}^{'T}$ through various extraction techniques.

### 3.5 EXTENSION POTENTIAL OF SUPERMARK

**Other image-conditioned models.** Since the SR model is a core component of our framework and can be easily swapped out, future improvements can leverage more advanced SR models to enhance performance. In Sec. 4.3, we will discuss how enhancing the SR model directly improves both the robustness and fidelity of SuperMark. Beyond super-resolution models, other image-conditioned diffusion models could also be explored, as long as the denoised and conditional images can be closely aligned. This opens up opportunities for further enhancing the framework's flexibility and performance.

**Different watermark injection methods.** Several existing works have focused on enhancing Gaussian Shading and Tree-Ring methods, or introducing novel techniques for watermark injection into Gaussian noise, such as Ring-ID (Ci et al., 2024) and DiffuseTrace (Lei et al., 2024). The robustness of SuperMark is significantly influenced by the watermark injection technique employed. In Sec. 4.3, we will explore how the watermark injection method utilized in Tree-Ring (Wen et al., 2024) enables SuperMark to exhibit exceptional resistance to geometric distortions, such as rotations. Therefore, these methods can be seamlessly integrated into our framework, leveraging their advantages to enhance the corresponding robustness of SuperMark.

**Inversion accuracy.** The robustness of SuperMark is closely tied to the accuracy of the Inversion process: improving Inversion accuracy can reduce the reconstruction error of the initial watermarked noise, thereby enhance the watermark extraction accuracy. Several existing works are exploring more precise Inversion techniques beyond the basic DDIM Inversion, such as those proposed in Hong et al. (2024) and Meiri et al. (2023). Integrating these advanced methods could further bolster the robustness of SuperMark.

**Inference overhead.** Since SuperMark requires multiple iterative steps of inference and inversion to achieve watermark embedding and extraction, it results in significant inference overhead. However, numerous efforts have been made to accelerate diffusion models, such as using more efficient

sampling methods (Salimans & Ho, 2022), model distillation (Meng et al., 2023), and consistency models (Song et al., 2023). These approaches can reduce the number of sampling steps to just a few, or even a single step, while maintaining image generation quality. For example, SinSR (Wang et al., 2024) is proposed recently to achieve single-step SR generation with a student model obtained by distillation. Additionally, SinSR has demonstrated improved Inversion accuracy, positioning it as another effective approach for enhancing the robustness of SuperMark. In the future, SuperMark can flexibly integrate these acceleration techniques to reduce time costs and enhance practicality.

## 4 EXPERIMENTS

### 4.1 EXPERIMENTAL SETTING

**Datasets.** For our evaluation, we use a default dataset consisting of 500 randomly selected images from the MS-COCO dataset (Lin et al., 2014), a large-scale real-world dataset containing 328K images. Since InstructPixPix requires paired instruction-image data, we extract 500 pairs from the official dataset [1] to assess robustness. Additionally, to further validate the effectiveness of Super-Mark, we conduct tests on several other datasets: DiffusionDB (Wang et al., 2023), WikiArt (Phillips & Mackintosh, 2011), CLIC (Toderici et al., 2020), and MetFACE (Karras et al., 2020), which are commonly used in RoSteALS (Bui et al., 2023) and ZoDiac Zhang et al. (2024) benchmarks. Specifically, we randomly select 500 images from DiffusionDB, WikiArt, and MetFACE, and use the entire test set of 428 images from CLIC. All images are resized and center-cropped to a resolution of $512 \times 512$.

**Implementation details.** We use the SD-Upscaler [2] as our default super-resolution (SR) model. For both sampling and inversion, the following configurations are applied: prompt = Null, guidance scale = 1.0, noise level = 0, and steps = 25. The low-resolution image size, $S_{low}$, is set to 128, and the strength factor, $f_s$, is set to 0.4. For watermark injection, we configure Gaussian Shading to embed 32 bits. To evaluate robustness, we consider the following normal distortions: JPEG compression, random cropping, Gaussian blur, Gaussian noise, and brightness adjustments. Additionally, we examine adaptive attacks, including VAE-based methods such as Bmshj18 (Ballé et al., 2018) and Cheng20 (Cheng et al., 2020), as well as diffusion-based attacks like Zhao23 (Zhao et al., 2023) and InstructPix2Pix (InsP2P) (Brooks et al., 2023). Detailed configurations are provided in Appendix A.2.

**Metrics.** For assessing the fidelity of watermarked images, we utilize Peak Signal-to-Noise Ratio (*PSNR*) and Structural Similarity Index Measure (*SSIM*). To measure the robustness and accuracy of watermark extraction, we use *Bit Accuracy*. This metric indicates the proportion of watermark bits correctly recovered during the extraction process, providing a direct measure of the watermarking method's effectiveness in preserving and retrieving the embedded information.

### 4.2 MAIN RESULTS

Table 1: Comparison results of SuperMark and baseline methods in terms of fidelity and watermark extraction ability. The **best** and the second best results are highlighted in bold and underlined, respectively.

| Method | Fidelity | | | Watermark Extraction Ability (Accuracy ↑) | | | | | | | | | | |
| | PSNR↑ | SSIM↑ | Identity | Normal Distortions | | | | | | Adaptive Attacks | | | | |
| | | | | JPEG | Crop | G Blur | G Noise | Brightness | Average | Bmshj18 | Cheng20 | Zhao23 | InsP2P | Average |
|---|---|---|---|---|---|---|---|---|---|---|---|---|---|---|
| DwtDct | 38.0227 | 0.9652 | 0.9214 | 0.5096 | 0.7881 | 0.5227 | 0.7022 | 0.5635 | 0.6172 | 0.5026 | 0.5027 | 0.5031 | 0.5011 | 0.5024 |
| DwtDctSvd | 38.1125 | 0.9730 | 0.9988 | 0.9623 | 0.8040 | 0.9917 | 0.8368 | 0.5691 | 0.8328 | 0.5060 | 0.5034 | 0.5006 | 0.4950 | 0.5013 |
| RivaGAN | 40.5255 | 0.9788 | 0.9988 | 0.9624 | 0.9967 | 0.9963 | 0.9088 | 0.9490 | 0.9626 | 0.5669 | 0.5618 | 0.6399 | 0.5905 | 0.5898 |
| StegaStamp | 28.6922 | 0.8957 | 0.9987 | 0.9981 | 0.9753 | 0.9958 | 0.9236 | 0.9657 | 0.9717 | **0.9979** | **0.9981** | **0.9260** | **0.9209** | **0.9607** |
| MBRS | **43.2538** | **0.9874** | **1.0000** | 0.9965 | 0.8605 | 0.7104 | 0.8035 | 0.9316 | 0.8605 | 0.5607 | 0.5547 | 0.5291 | 0.5296 | 0.5435 |
| CIN | 41.7388 | 0.9789 | **1.0000** | 0.6274 | **1.0000** | 0.9130 | 0.9178 | **0.9966** | 0.8910 | 0.5084 | 0.5128 | 0.5026 | 0.5020 | 0.5065 |
| PIMoG | 37.4647 | 0.9772 | 0.9989 | 0.7804 | 0.9918 | 0.9871 | 0.7078 | 0.9191 | 0.8772 | 0.6336 | 0.6015 | 0.5483 | 0.5191 | 0.5756 |
| SepMark | 35.9085 | 0.9520 | 0.9997 | **0.9985** | 0.9932 | 0.9889 | 0.9667 | 0.9760 | 0.9847 | 0.8312 | 0.8511 | 0.7466 | 0.7394 | 0.7921 |
| RoSteALS | 28.3445 | 0.8396 | 0.9947 | 0.9745 | 0.8500 | 0.9908 | 0.9231 | 0.9412 | 0.9359 | 0.9074 | 0.9034 | 0.8469 | 0.8412 | 0.8747 |
| SuperMark | 32.4978 | 0.9322 | **1.0000** | 0.9976 | **1.0000** | **1.0000** | **0.9810** | 0.9946 | **0.9946** | 0.9293 | 0.9310 | 0.8718 | 0.8396 | 0.8929 |

---

[1]https://huggingface.co/datasets/timbrooks/instructpix2pix-clip-filtered

[2]https://huggingface.co/stabilityai/stable-diffusion-x4-upscaler

We compare SuperMark with nine open-source baselines, and the results are presented in Table 1. The watermarked images generated by SuperMark exhibit relatively high fidelity, comparable to other baselines. Notably, SuperMark demonstrates significantly stronger robustness than most of the watermarking methods tested. Against normal distortions, SuperMark achieves the highest average watermark extraction accuracy of 99.46%. Even in the face of adaptive attacks, which render most watermarking methods ineffective, SuperMark maintains a high robustness with an accuracy of 89.29%. Although this accuracy is slightly lower than StegaStamp, which may have state-of-the-art robustness, SuperMark outperforms in terms of fidelity, with visual results and analyses provided in Appendix A.3.

## 4.3 Transferability

**Transfer to different datasets.** To evaluate the universality of SuperMark across data with different distributions, we conduct additional experiments on four datasets: DiffusionDB, WikiArt, CLIC, and MetFACE. As shown in Table 2, SuperMark performs effectively across these diverse data distributions. Notably, in the MetFACE dataset, watermarked images exhibit superior fidelity, particularly in terms of PSNR. This may be due to the SR model's proficiency in enhancing details for facial images, allowing

Table 2: Test results of SuperMark on different datasets. The gray cell denotes the default setting.

| Dataset | Fidelity | | Watermark Extraction Ability↑ | | |
|---|---|---|---|---|---|
| | PSNR↑ | SSIM↑ | Identity | Normal Distortions | Adaptive Attacks |
| DiffusionDB | 32.5958 | 0.9318 | 1.0000 | 0.9942 | 0.8751 |
| WikiArt | 32.1425 | 0.9126 | 1.0000 | 0.9950 | 0.9064 |
| CLIC | 33.0314 | 0.9387 | 1.0000 | 0.9939 | 0.8870 |
| MetFACE | 37.2351 | 0.9363 | 1.0000 | 0.9952 | 0.8330 |
| COCO | 32.4978 | 0.9322 | 1.0000 | 0.9946 | 0.8929 |

the generated images to closely approximate the originals. These results further support the idea that the stronger the SR model, the higher the fidelity achieved. Visualizations of the results for different datasets are provided in Appendix A.3.

**LDM-SR as the SR model.** Given the flexible selection of the SR model in SuperMark, we also test it with another SR model, LDM-SR [3] for transferability assessment, to demonstrate SuperMark's versatility. As the VAE used in LDM-SR has 3 latent channels, it is not possible to configure a 32-bit watermark. To ensure a fair comparison, we use 16 embedding bits for both super-resolution models ($f_c = 3$ for LDM-SR and $f_c = 4$ for SD-Upscaler). As shown in Table 3, SuperMark with LDM-SR achieves comparable performance in both fidelity and robustness against normal distortions. However, SD-Upscaler, an SR model with superior performance compared to LDM-SR, may provide SuperMark with greater robustness against adaptive attacks. This confirms that improving the capabilities of the SR model used in SuperMark can enhance its overall robustness. We also provide some visual examples in Appendix A.3.

Table 3: Test results of SuperMark using different SR models.

| SR Model | Fidelity | | Watermark Extraction Ability↑ | | | | | | | | | | | |
|---|---|---|---|---|---|---|---|---|---|---|---|---|---|---|
| | PSNR↑ | SSIM↑ | Identity | Normal Distortions | | | | | | Adaptive Attacks | | | | |
| | | | | JPEG | Crop | G Blur | G Noise | Brightness | Average | Bmshj18 | Cheng20 | Zhao23 | InsP2P | Average |
| LDM-SR | 32.3906 | 0.9332 | 1.0000 | 0.9972 | 1.0000 | 1.0000 | 0.9632 | 0.9930 | 0.9907 | 0.9300 | 0.9297 | 0.9411 | 0.8716 | 0.9181 |
| SD-Upscaler | 32.4747 | 0.9306 | 1.0000 | 0.9998 | 1.0000 | 1.0000 | 0.9883 | 0.9945 | 0.9965 | 0.9626 | 0.9628 | 0.9511 | 0.9066 | 0.9458 |

**Adoption of Tree-Ring's watermark injection method.** We also utilize the watermark injection method employed in Tree-Ring (Wen et al., 2024) to further assess the transferability of SuperMark. The configurations of Tree-Ring are: the watermark ring radius $r$ is set to 30 and the threshold $\tau$ is set to 0.9, which means the watermark is detected if $p$ falls below this value. As ZoDiac is not open source, we apply the same distortions and attack configurations as described in their paper for comparison (see Appendix A.2 for details). Since Tree-Ring is a 0-bit watermark method, we use the Watermark Detection Rate (WDR) to evaluate the performance aligned with ZoDiac.

The test results of applying Tree-Ring to SuperMark are presented in Table 4 and we also provide some visual results in Appendix A.3. Due to the ring-shaped watermark embedded with the Tree-Ring method, SuperMark demonstrates superior robustness against spatial distortions, such as rotation. Furthermore, compared to ZoDiac, SuperMark maintains better fidelity and exhibits sig-

---

[3]https://huggingface.co/CompVis/ldm-super-resolution-4x-openimages

nificantly stronger robustness against rotation, while offering comparable robustness against other distortions and attacks.

Table 4: Comparison results of ZoDiac and SuperMark adopting Tree-Ring's watermark injection method. The corresponding results of ZoDiac are those presented in their paper.

| Method | Fidelity | | Watermark Extraction Ability↑ | | | | | | | | | | |
|---|---|---|---|---|---|---|---|---|---|---|---|---|---|
| | PSNR↑ | SSIM↑ | Identity | Normal Distortions | | | | | | Adaptive Attacks | | | |
| | | | | JPEG | G Blur | G Noise | Brightness | Rotation | Average | Bmshj18 | Cheng20 | Zhao23 | Average |
| ZoDiac | 29.41 | 0.92 | 0.998 | 0.992 | 0.996 | 0.996 | 0.998 | 0.538 | 0.904 | 0.992 | 0.986 | 0.988 | 0.989 |
| SuperMark | 32.65 | 0.94 | 1.000 | 0.998 | 1.000 | 0.962 | 0.998 | 0.978 | 0.987 | 0.968 | 0.990 | 0.952 | 0.970 |

**Transfer to different resolutions.** Watermarks can be injected into Gaussian noise of varying sizes, enabling SuperMark to embed and extract watermarks for images of different resolutions. As higher-resolution images offer more capacity for watermark embedding, it becomes feasible to embed more bits. To maintain consistency, we keep $f_c$ and $f_{hw}$ constant, ensuring the same copy count of each bit, which will also change the length of the embedded bits. Besides, we maintain $f_s$ unchanged to

Table 5: SuperMark's test results on images of different resolutions.

| Resolution | Bits | Fidelity | | Watermark Extraction Ability↑ | | |
|---|---|---|---|---|---|---|
| | | PSNR↑ | SSIM↑ | Identity | Normal Distortions | Adaptive Attacks |
| 256 | 8 | 29.5828 | 0.8929 | 1.0000 | 0.9963 | 0.9091 |
| 384 | 18 | 31.5776 | 0.9239 | 1.0000 | 0.9957 | 0.9014 |
| 512 | 32 | 32.4978 | 0.9322 | 1.0000 | 0.9946 | 0.8929 |
| 640 | 50 | 33.6769 | 0.9392 | 1.0000 | 0.9936 | 0.9002 |
| 768 | 72 | 34.5080 | 0.9408 | 1.0000 | 0.9938 | 0.9012 |

control the watermark strength added to the original image. This setup allows us to evaluate the impact of resolution on SuperMark's performance.

The results, shown in Table 5, indicate that SuperMark improves fidelity as image resolution increases while maintaining robustness against both normal distortions and adaptive attacks with more bits embedded. This improvement is due to SR models being more effective at upscaling higher-resolution images, whereas upscaling lower-resolution images requires adding more details and involves a larger generative space, which is more challenging. As a result, the generated high-resolution image differs more significantly from the corresponding watermarked image, leading to lower fidelity. This also suggests that enhancing the SR model's capabilities can improve the fidelity.

## 4.4 ABLATION STUDY

**Impact of low image size $S_{low}$ and strength factor $f_s$.** Two important hyperparameters, $S_{low}$ and $f_s$, play a key role in balancing the fidelity of watermarked images and the bit accuracy of watermark extraction. We conduct a series of comprehensive experiments to explore different combinations of these parameters, and the results are displayed in Figure 3. When $f_s$ is fixed, increasing $S_{low}$ enhances the fidelity but reduces robustness. This is consistent with our previous analysis: larger $S_{low}$ leads to fewer pixel losses in the original image during watermark embedding, but more pixel losses in the watermarked image during extraction. Moreover, for a given $S_{low}$, increasing $f_s$, which amplifies the strength of the added watermark residual, improves the robustness of the watermark. It is also worth noting that smaller $S_{low}$ values reduce the memory and time overhead required for both inference and inversion. In practical use, we can configure $S_{low}$ and $f_s$ to maintain both fidelity and robustneelatively high levels. By default, we set $S_{low} = 128$ and $f_s = 0.4$, as this reduces memory usage during inference and improves inference speed.

**Impact of inference and inversion steps.** We evaluate SuperMark's performance under varying inference and inversion steps, with results presented in Figure 4. Our observations show that different inversion steps have minimal impact on both the fidelity and robustness of SuperMark. However, increasing the number of inference steps results in a slight decrease in fidelity while significantly improving robustness, especially in scenarios involving adaptive attacks. We hypothesize that more inference steps prompt the SR model to generate more detailed features, which increases the discrepancy between the watermarked and original images, leading to a decline in fidelity. Conversely, these generated details provide more reversible pixels during the inversion process, thereby enhancing watermark extraction accuracy.

**Impact of watermark bits length.** We can embed bits of varying lengths by adjusting different values of $f_c$ and $f_{hw}$ and the results are shown in Figure 5. The fidelity of the watermarked image is

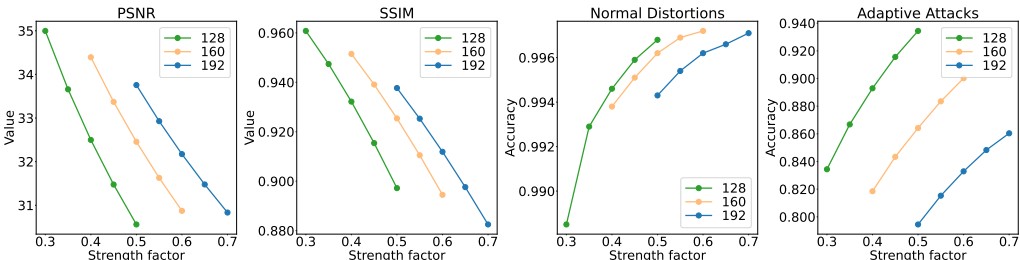

Figure 3: The impact of varying the low image size $S_{low}$ and strength factor $f_s$ on the fidelity and robustness. Robustness is measured by the watermark extraction accuracy on watermarked images subjected to normal distortions and adaptive attacks. Lines of different colors represent different values of $S_{low}$.

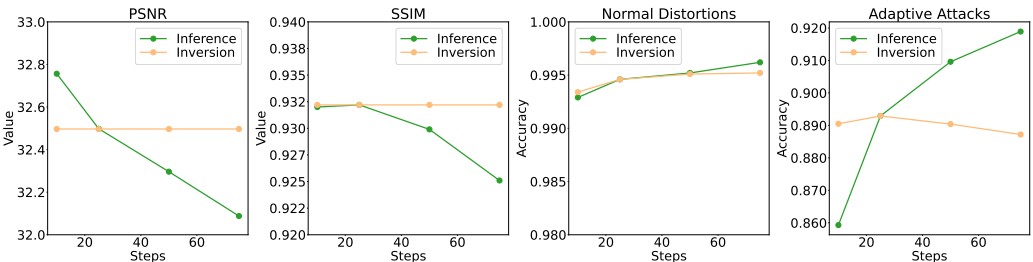

Figure 4: Effects of SuperMark on fidelity and robustness with varying inference and inversion steps.

maintained across different bit lengths, as it has been shown that in Gaussian Shading, the sampling of Gaussian noise based on bits does not affect the model's denoising performance. Consequently, the SR model generates images with consistent fidelity, irrespective of the bit length employed. However, embedding more bits leads to a corresponding decrease in SuperMark's robustness, particularly when faced with adaptive attacks. This is expected, as embedding more bits requires a larger number of successfully inverted pixels for extraction, while the proportion of invertible pixels in a corrupted image remains fixed. Consequently, the accuracy of watermark extraction diminishes as the bit length increases.

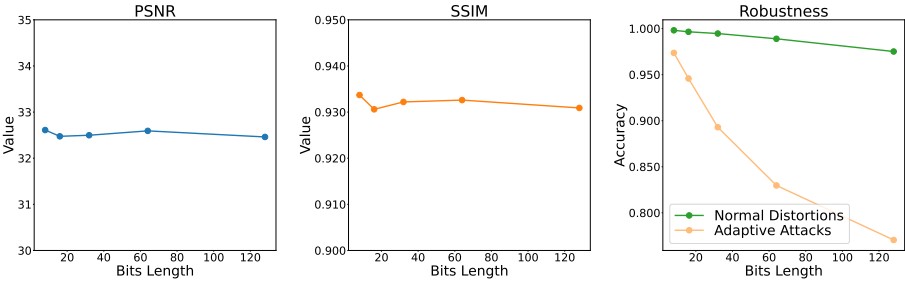

Figure 5: Fidelity and robustness when embedding watermark bits of different lengths.

## 5 CONCLUSION

In this paper, we propose a training-free and robust image watermarking framework, named SuperMark, which leverages a diffusion-based SR model to achieve effective watermark embedding and extraction. Thanks to the inherent resilience of DDIM Inversion to various distortions, SuperMark demonstrates superior robustness compared to nearly all existing watermarking methods while maintaining high fidelity. Extensive experiments highlight its outstanding performance across different datasets, watermark injection methods, SR models, and image resolutions.

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

# A  APPENDIX

## A.1  PRELIMINARY

### A.1.1  GAUSSIAN SHADING

The watermark is a bit string $s$ consisting of 0s and 1s, with a length defined as $\frac{c}{f_c} \cdot \frac{h}{f_{hw}} \cdot \frac{w}{f_{hw}}$, where $c$, $h$, and $w$ represent the channels, height, and width of the Gaussian noise used for watermark injection, and $f_c$, $f_{hw}$ are scaling factors for expansion. The string $s$ is then replicated $f_c \cdot f_{hw}^2$ times and reshaped into its diffused version $s^d$ with the shape $(c, h, w)$. To preserve the distribution and obtain the corresponding watermarked Gaussian noise $Z_{wm}^T$, $s^d$ is transformed into a uniformly distributed randomized watermark $m$ through encryption (e.g., ChaCha20 (Bernstein et al., 2008)) using a stream key $K$. The watermarked Gaussian noise $Z_{wm}^T$ is sampled as follows:

$$p(Z_{wm}^T|y = i) = \begin{cases} 2 \cdot f(Z_{wm}^T) & ppf\left(\frac{i}{2}\right) < Z_{wm}^T \leq ppf\left(\frac{i+1}{2}\right) \\ 0 & \text{otherwise} \end{cases},$$

where $y \in \{0, 1\}$ is the bit in $s^d$. Since $m$ follows a uniform distribution, it can be shown that $Z_{wm}^T$ preserves the Gaussian distribution, ensuring that the fidelity of the image denoised from $Z_{wm}^T$ is not affected.

After performing DDIM inversion, the inverted Gaussian noise $Z_{wm}^{'T}$ is obtained, and the diffused watermark $s^{'d}$ is extracted by:

$$i^{'} = \lfloor 2 \cdot cdf(Z_{wm}^{'T}) \rfloor,$$

where $i^{'}$ is the extracted bit in $m^{'}$. The decrypted version of $m^{'}$ using $K$ yields $s^{'d}$, which consists of $f_c \cdot f_{hw}^2$ copies of the watermark. The extracted watermark $s^{'}$ is then reconstructed using a voting mechanism: if a bit is set to 1 in more than half of the copies, the corresponding bit in $s^{'}$ is set to 1; otherwise, it is set to 0.

### A.1.2 TREE-RING

The watermark is a key $k^*$ composed of multiple rings, with a constant value along each ring. The key $k^*$ is injected into the Fourier transform of the initial Gaussian noise $Z^T$ to obtain the watermarked Gaussian noise $Z^T_{wm}$. Specifically, a circular mask $M$ with radius $r$ centered on the low-frequency modes is chosen, and the injection process is described as:

$$\mathcal{F}(Z^T_{wm}) \sim \begin{cases} k^*_i & i \in M \\ \mathcal{N}(0,1) & \text{otherwise} \end{cases},$$

For watermark extraction, let $y = \mathcal{F}(Z'^T_{wm})$, and the score $\mu$ is defined as:

$$\mu = \frac{1}{\sigma^2} \sum_{i \in M} |k^*_i - y|^2,$$

where $\sigma^2 = \frac{1}{M} \sum_{i \in M} |y_i|^2$. An interpretable P-value $p$ is computed as:

$$p = \Pr(\chi^2_{|M|,\lambda} \le \mu \mid H_0) = \Phi_{\chi^2}(z),$$

where $\Phi_{\chi^2}(z)$ is a standard statistical function. The watermark is "detected" when $p$ falls below a chosen threshold $\alpha$.

### A.2 MORE IMPLEMENTATION DETAILS

**Configurations of normal distortions.** The default configurations of different normal distortions are: JPEG (Q=50), Random Crop (ratio=0.8), Gaussian Blur (r=2), Gaussian Noise (std=0.05), Brightness (factor=2). When testing on Tree-Ring, the configurations are: JPEG (Q=50), Gaussian Blur (r=5), Gaussian Noise (std=0.05), Brightness (factor=0.5), Rotation (degrees=90).

**Configurations of adaptive attacks.** For Bmshj18 and Cheng20, we use the models from CompressAI [4] (bmshj2018_hyperprior and cheng2020_anchor) with compression factor=3. For Zhao23, we use the model [5] with noise&denoise steps=20 by default and steps=60 when testing on Tree-Ring. For InstructPix2Pix, we use the model [6] with text guidance=7.5, image guidance=1.5 and inference steps=25.

### A.3 VISUAL RESULTS

**Fidelity comparison with StegaStamp, RoSteALS and SuperMark.** Figure 7 compares the fidelity distribution of StegaStamp, RoSteALS and SuperMark, which have comparable robustness. SuperMark demonstrates the best performance in both PSNR and SSIM, with stability only slightly behind StegaStamp. While StegaStamp shows relatively consistent results, its fidelity lags behind SuperMark. On the other hand, RoSteALS exhibits significant variability in both PSNR and SSIM, resulting in lower and less stable fidelity. Figure 8 showcases some examples where SuperMark produces relatively low fidelity, primarily due to the complex composition and detailed content of the original images. The SR model adopted in SuperMark may face challenges with these intricate contents and have more generative freedom, leading to lower fidelity. However, we believe that future advancements in more powerful SR models will further enhance the fidelity for such images.

We also select some watermarked images generated by StegaStamp, RoSteALS and SuperMark which can be found in Figure 9 and Figure 10. From the residual images, we can see that the watermarks embedded by our method are more concentrated at the edges of objects, that is, at places with strong semantic correlation, thus ensuring both fidelity and strong robustness. However, StegaStamp embeds more watermarks in both objects and backgrounds, which improves robustness but sacrifices more fidelity.

---

[4] https://github.com/InterDigitalInc/CompressAI/tree/master

[5] https://huggingface.co/stable-diffusion-v1-5/stable-diffusion-v1-5

[6] https://huggingface.co/timbrooks/instruct-pix2pix

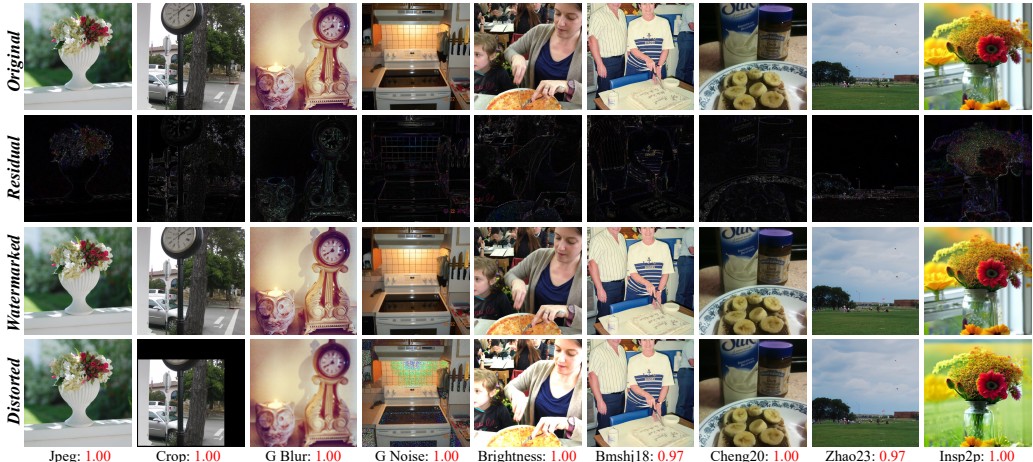

Figure 6: Some visual results from the default COCO dataset. The last row marks the distortion or attack type of each column and the corresponding watermark extraction accuracy.

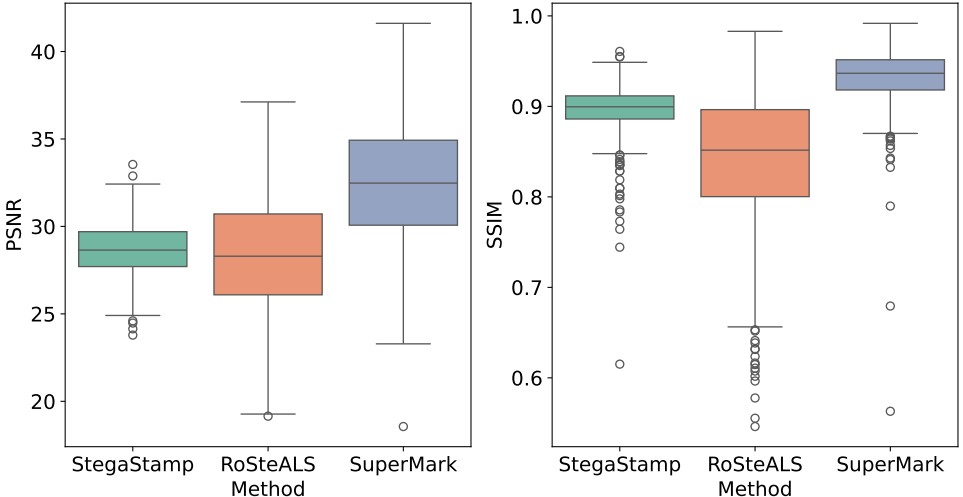

Figure 7: Fidelity distribution of watermarked images generated by StgeaStamp, RoSteALS and SuperMark on the default COCO dataset.

**Watermarked images generated by SuperMark on different datasets.** See Figure 11, Figure 12, Figure 13 and Figure 14. It can be observed that for different types of images from various datasets, SuperMark is able to achieve high-fidelity watermarked image generation, with the watermark embedded at the edges of semantically relevant objects.

**Watermarked images generated by SuperMark using LDM-SR as the SR model and Tree-Ring as the watermark injection method.** See Figure 15 and Figure 16. It can be observed that, despite using different SR models and watermark injection methods, SuperMark consistently shows similar embedding patterns on the same original image, leading to comparable fidelity. This further reinforces the strong transferability of our method.

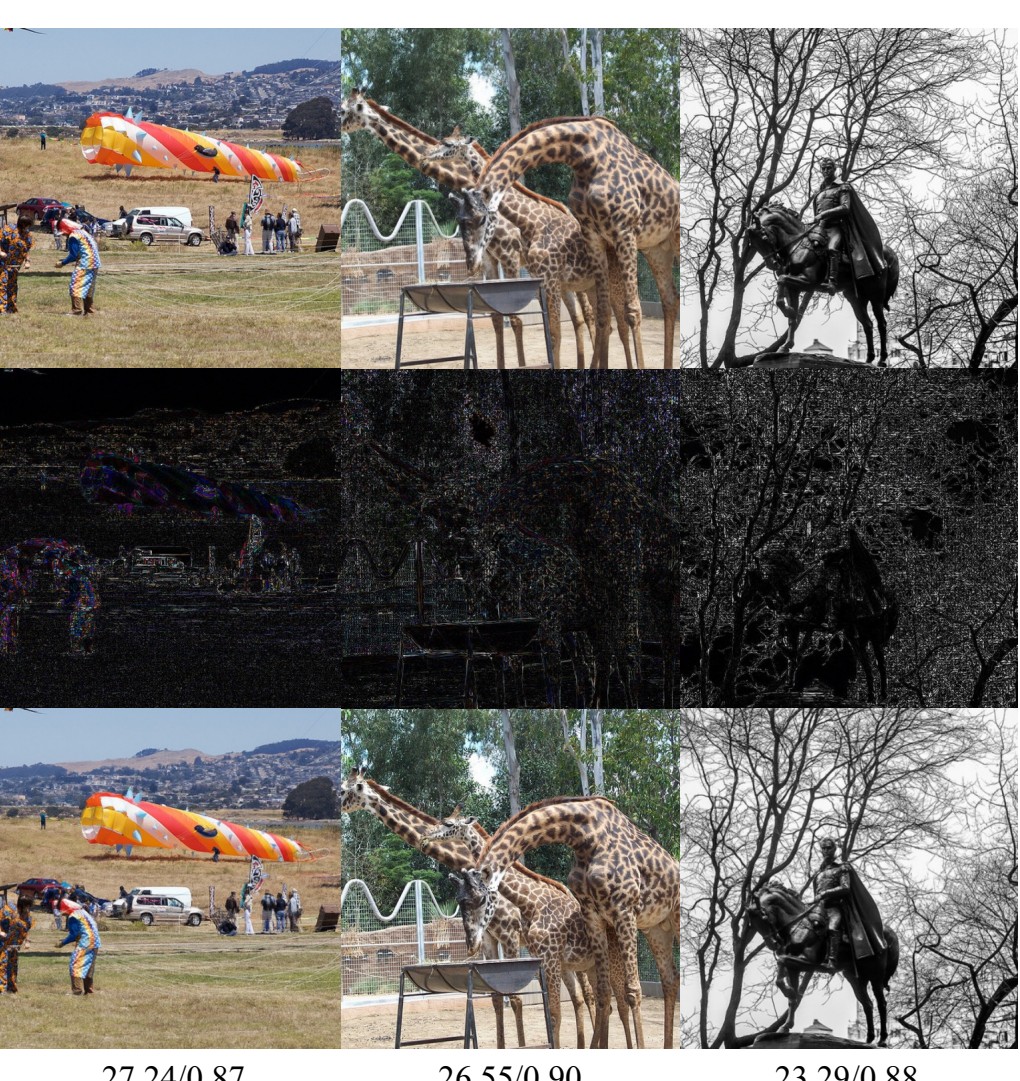

27.24/0.87          26.55/0.90          23.29/0.88

Figure 8: Some watermarked images with relatively low fidelity generated by SuperMark. From the first row to the fourth row are: original image, residual image, watermarked image and PSNR/SSIM. Same for the following Figures.

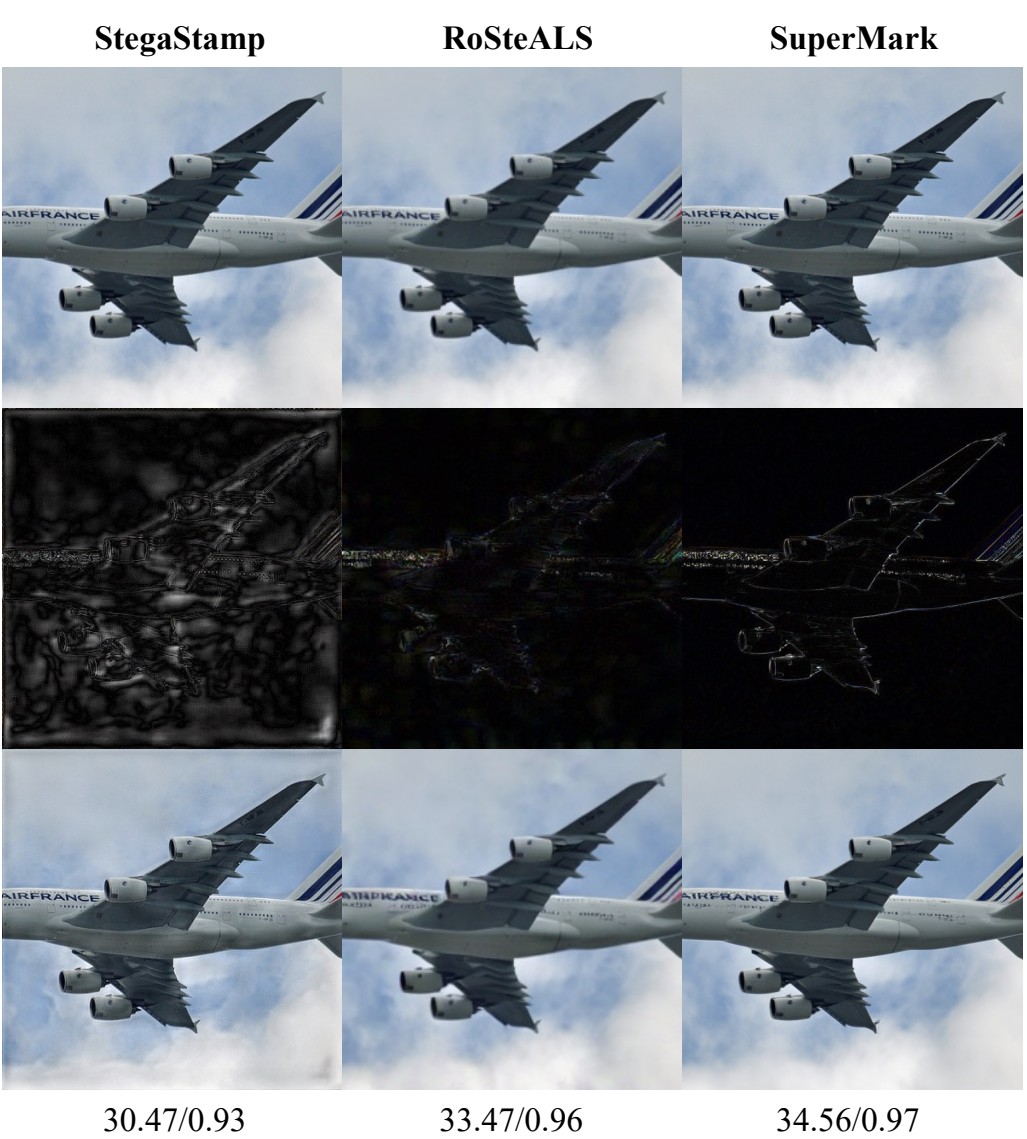

**StegaStamp**     **RoSteALS**     **SuperMark**

30.47/0.93     33.47/0.96     34.56/0.97

Figure 9: Comparison of watermarked images generated by StegaStamp, RoSteALS and Super-Mark.

| StegaStamp | RoSteALS | SuperMark |
|---|---|---|

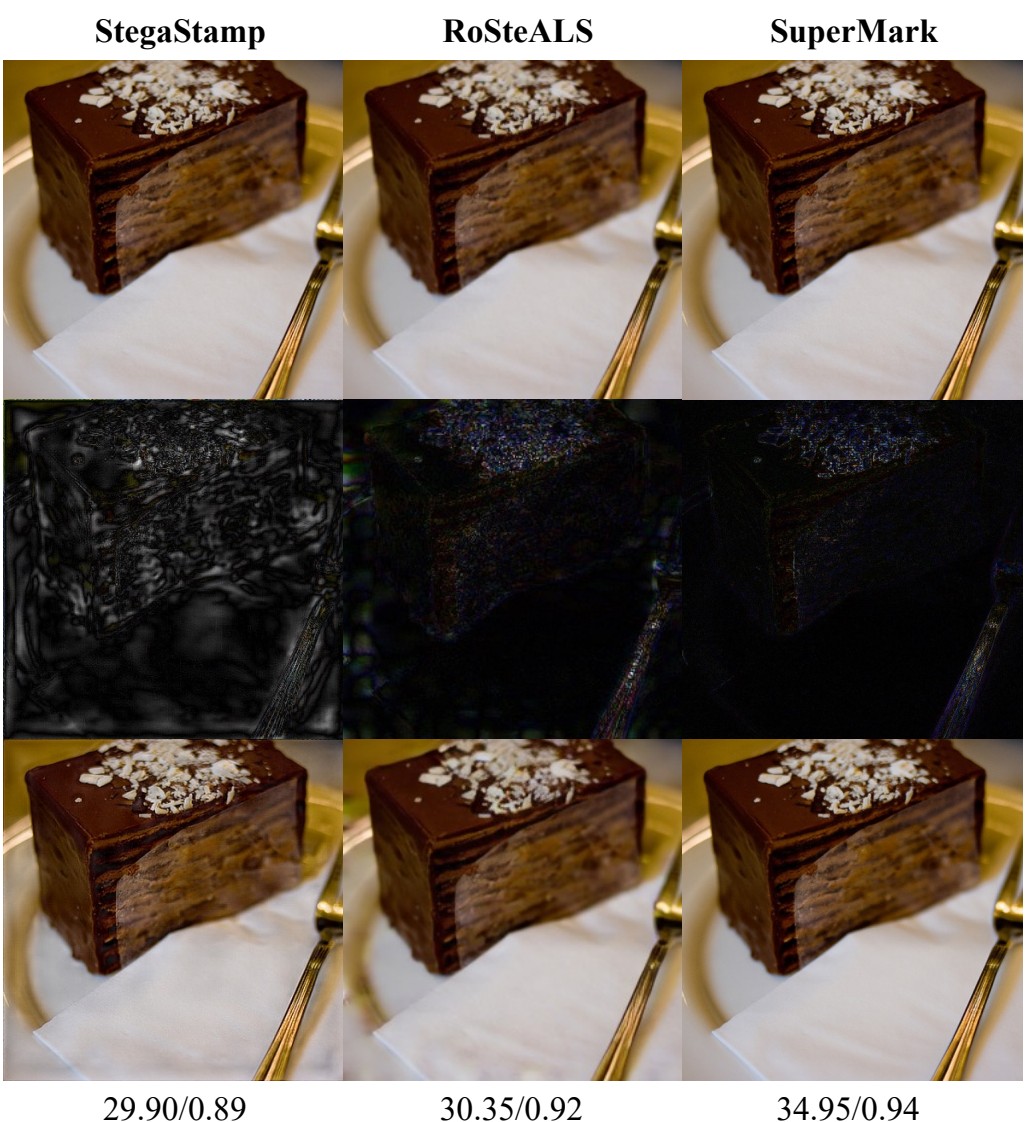

| 29.90/0.89 | 30.35/0.92 | 34.95/0.94 |
|---|---|---|

Figure 10: Comparison of watermarked images generated by StegaStamp, RoSteALS and Super-Mark.

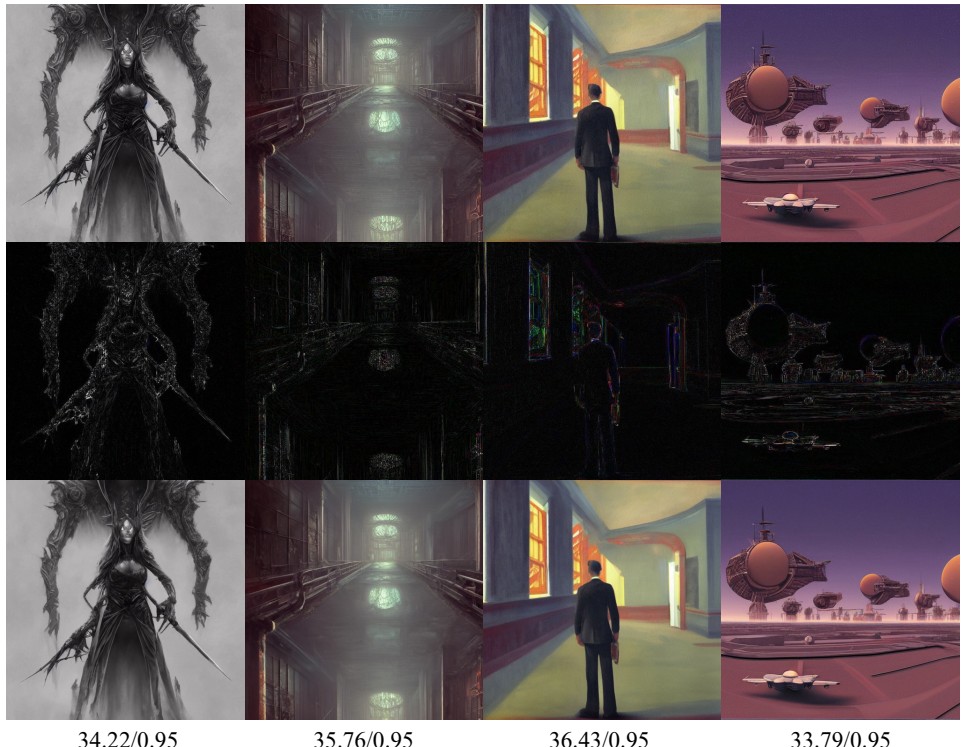

34.22/0.95   35.76/0.95   36.43/0.95   33.79/0.95

Figure 11: Some watermarked images generated by SuperMark with the original images sampled from the DiffusionDB dataset.

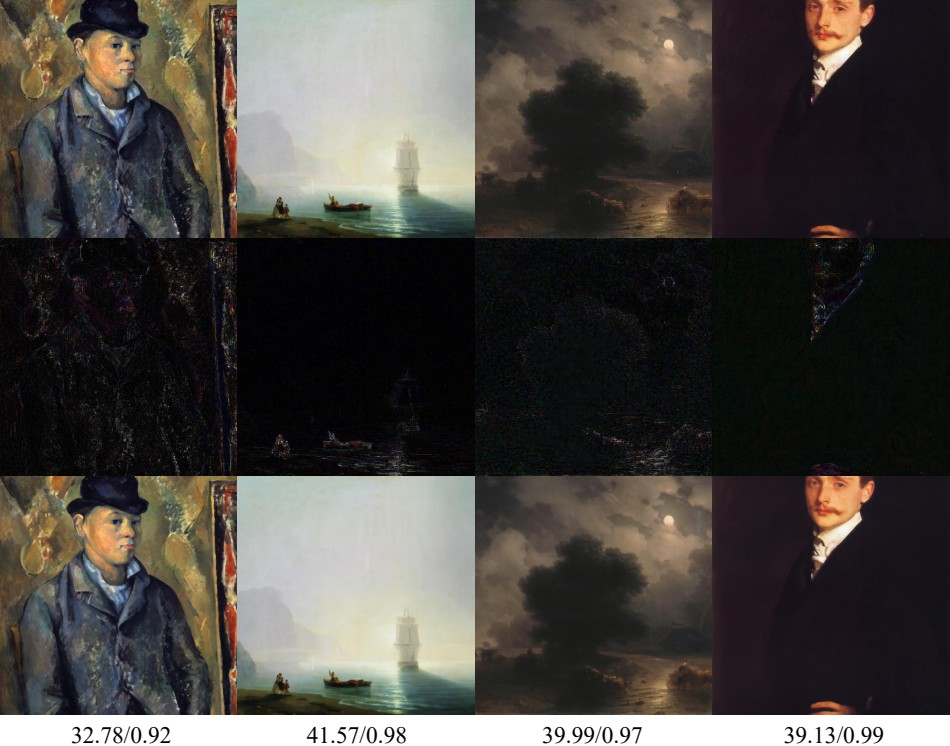

32.78/0.92   41.57/0.98   39.99/0.97   39.13/0.99

Figure 12: Some watermarked images generated by SuperMark with the original images sampled from the WikiArt dataset.

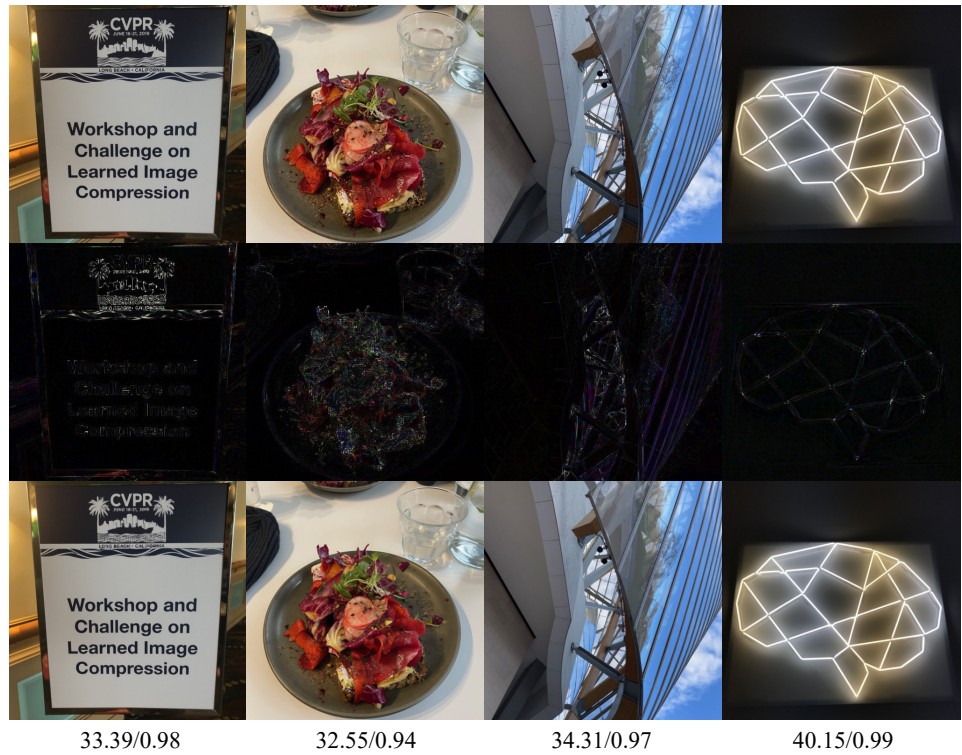

33.39/0.98        32.55/0.94        34.31/0.97        40.15/0.99

Figure 13: Some watermarked images generated by SuperMark with the original images sampled from the CLIC dataset.

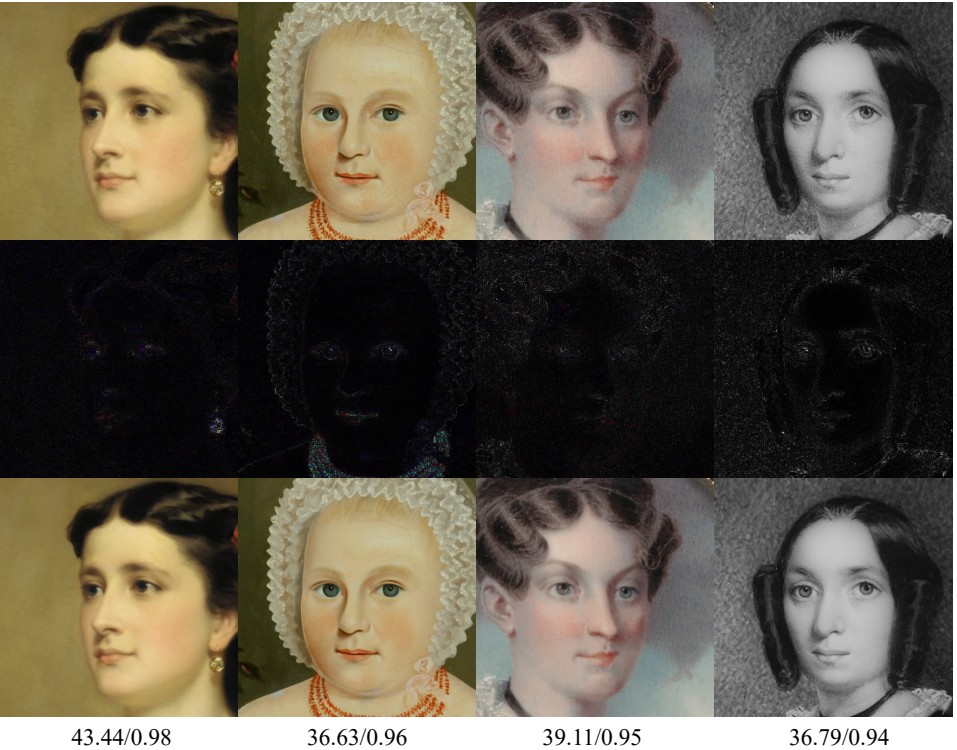

43.44/0.98        36.63/0.96        39.11/0.95        36.79/0.94

Figure 14: Some watermarked images generated by SuperMark with the original images sampled from the MetFACE dataset.

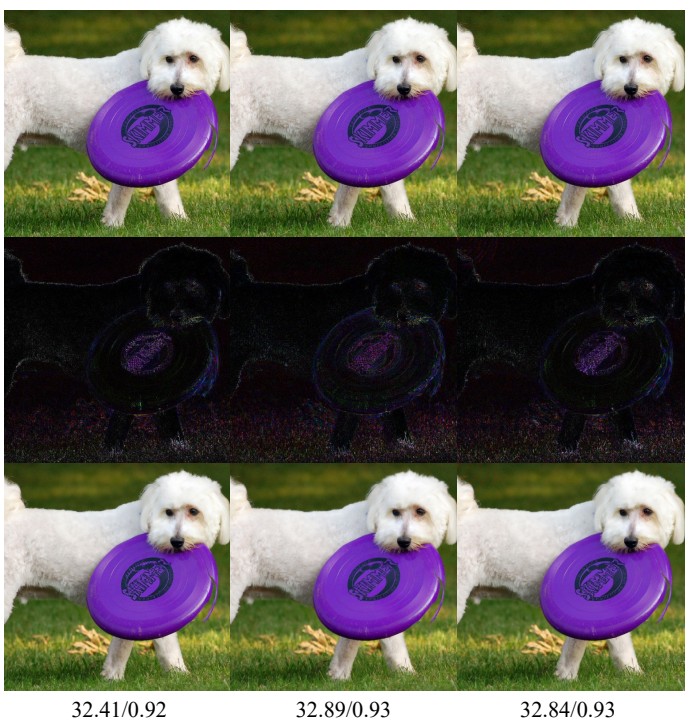

34.72/0.93         34.58/0.95         34.90/0.94

Figure 15: Comparison of watermarked images generated by SuperMark with default setting, LDM-SR as the SR model and Tree-Ring as the watermark injection method. The first column is the default setting, the second column is using LDM-SR, and the third column is using Tree-Ring. Same as Figure 16.

32.41/0.92         32.89/0.93         32.84/0.93

Figure 16: Comparison of watermarked images generated by SuperMark with default setting, LDM-SR as the SR model and Tree-Ring as the watermark injection method.

