# OpenReview forum: "SuperMark: Robust and Training-free Image Watermarking via Diffusion-based Super-Resolution"
_ICLR.cc/2025/Conference — Submitted to ICLR 2025_

### Official Review · Reviewer_e4Ga · 2024-11-01

**Soundness:** 2
**Presentation:** 3
**Contribution:** 2
**Rating:** 3
**Confidence:** 5

**Summary:**

This paper introduces SuperMark, a robust and training-free framework for watermarking AI-generated content to protect copyright and authenticate content. SuperMark leverages parallels between watermark embedding/extraction and the denoising processes in diffusion models, embedding watermarks into Gaussian noise and using SR models to recover them. Unlike traditional methods that struggle with robustness and fidelity, SuperMark demonstrates strong resilience against distortions, adaptive attacks, and maintains high fidelity through DDIM Inversion techniques. It achieves high watermark extraction accuracy (over 89%) across different conditions, surpassing many existing methods. Additionally, SuperMark shows strong adaptability across various datasets, SR models, and watermark settings.

**Strengths:**

- The paper presents a novel discovery regarding the diffusion process, highlighting its intrinsic robustness against normal distortions and adaptive attacks, which makes this finding particularly interesting.
- The diffusion process does not require training.
- The pipeline depicted in Figure 2 is clearly illustrated and facilitates easy comprehension.

**Weaknesses:**

- The experimental results do not SOTA performance, particularly in terms of PSNR and SSIM metrics. For instance, in Table 1, the PSNR value is only 32.5, which is considerably low, severely affecting image fidelity. This suggests that the trade-off between fidelity and robustness in your method does not yield a compelling outcome. Furthermore, the PSNR values reported in Table 2 remain significantly below current SOTA.

- The novelty of the work appears insufficient, as the methods employed seem to rely primarily on off-the-shelf algorithms. The primary contribution appears to be the strategic integration of these existing techniques.

- The paper lacks an analysis of efficiency. Although the diffusion process is advantageous as a training-free approach to counter the two types of attacks you mentioned, it appears to have drawbacks related to inference efficiency. The inference time seems potentially much lower compared to conventional encoder-noise-decoder frameworks.

- Additionally, the paper does not include experiments against some SOTA attack methods. For example, it is missing an evaluation using attacks[r1].

[r1] Evading Watermark Based Detection of AI-generated Content (CCS 2023).

- There is also an insufficient ablation study. I am particularly curious about the impact on SuperMark if the SR model is removed from the process.

**Questions:**

1. The PSNR value in Table 1 is only 32.5, which severely impacts image fidelity. How do you explain the lack of a compelling trade-off between fidelity and robustness in your method?

2. Compared to the current SOTA, the reported PSNR and SSIM values are significantly lower across multiple experiments. What do you identify as the primary performance bottleneck?

3. There is no analysis of inference efficiency in your work. How does the inference time of the diffusion process compare to conventional encoder-noise-decoder frameworks.

---

### Official Review · Reviewer_zLG7 · 2024-11-02

**Soundness:** 2
**Presentation:** 2
**Contribution:** 2
**Rating:** 3
**Confidence:** 5

**Summary:**

This paper proposes a novel robust and training-free watermarking framework, named SuperMark. It embeds the watermark into initial Gaussian noise using existing techniques and then applies pretrained Super-Resolution (SR) models to denoise the watermarked noise, producing the watermarked image. Experiments demonstrate SuperMark's effectiveness.

**Strengths:**

The watermarking framework is training-free.
An end-to-end pipeline can facilitate the training.
Experiments show its advantages.

**Weaknesses:**

The paper is written and organized poorly. The abstract is too lengthy, and not concise. The paper has limited novelties and only ensembles existing techniques. Neither a novel algorithm nor a newly constructed module are presented. It is not clear how to seamlessly integrate with various watermark injection methods and pre-trained diffusion-based SR models. The layout is not well, for example, the text font size in all equations looks smaller than that of the variable of the main text. Also, there is no theoretical contribution throughout the paper. The difference from Gaussian shading is missing, which is an important reference in the field.

**Questions:**

Only a combination of existing techniques can be seen in the paper.

Diffusion can generate directly large-resolution images, so why need an SR module?

SR may impair the extraction of watermarks.

---

### Official Review · Reviewer_iJ1C · 2024-11-02

**Soundness:** 3
**Presentation:** 2
**Contribution:** 1
**Rating:** 3
**Confidence:** 4

**Summary:**

This paper proposes a post-processing watermarking scheme based on diffusion-based super resolution model. The author attempts to use the inherent robustness of stable diffusion to various distortions and presents a training-free framework SuperMark. SuperMark embeds the watermark into initial Gaussian noise and then applies pretrained super resolution models to denoise the watermarked noise. Experiments demonstrates its efficacy.

**Strengths:**

-	The proposed method is training-free and has lower time and computing cost than other watermarking methods.

**Weaknesses:**

-	This paper uses several overly arbitrary expressions. E.g., in Introduction, “The latter approach is more flexible and general, as it can be applied to both AIGC and real images”. In fact, recent AIGC watermarking research focuses more on in-process watermarking [1][2] because post-processing may be bypassed by malicious users. We believe that SuperMark can be regarded as a new attempt at real-image watermarking, and there is no need to force a connection with AIGC. Moreover, like “diffusion process holds inherent robustness against different distortions” in line 81. We believe that this robustness only applies to stable diffusion, not all diffusion methods. In fact, the experiment in this paper also uses a super resolution method based on stable diffusion.
-	The use of super-resolution in this model is completely unnecessary. Considering the whole process as illustrated in Figure 2, the image in the transmission is still the normal resolution image, not the higher/lower resolution image. In conclusion, we believe that the proposed method is just trying to introduce the diffusion process into the framework, and it does not matter for which task this model is designed. It seriously weakens the contribution of the paper since the performance is mostly brought by diffusion itself, but not any novel design from the proposed method.
-	Performance was just mentioned above, and in fact the performance of this paper is not very good. Both visual quality and decoding accuracy cannot achieve the best (as shown in Table 1).
-	The paper also contains some parts that are not nice or clear enough. For example, the double-headed arrow between the two "m" in Figure 1 does not make it clear what it means. We assume that this means that the original information and the extracted information need to be as similar as possible, so why is there no double-headed arrow between the original image and the embedded image? In addition, Figure 2 also lacks the necessary legend, which affects the readability of the paper.

[1] Wen, Yuxin, et al. "Tree-rings watermarks: Invisible fingerprints for diffusion images." Advances in Neural Information Processing Systems 36 (2024).

[2] Fernandez, Pierre, et al. "The stable signature: Rooting watermarks in latent diffusion models." Proceedings of the IEEE/CVF International Conference on Computer Vision. 2023.

**Questions:**

NA

---

### Official Review · Reviewer_RvRL · 2024-11-03

**Soundness:** 3
**Presentation:** 2
**Contribution:** 2
**Rating:** 3
**Confidence:** 4

**Summary:**

This paper introduces a post-processing watermarking scheme that leverages a diffusion-based super-resolution model. The author aims to exploit the inherent resilience of stable diffusion to withstand various distortions, resulting in a training-free framework. In this approach, the watermark is embedded within initial Gaussian noise, which is subsequently processed using pretrained super-resolution models to denoise the watermark.

**Strengths:**

-	The paper clearly points out the parts of the proposed framework that come from other works, and this candor could be a strength of the paper.

**Weaknesses:**

- My humble thought on this paper is that the method is very similar to a “super-resolution-variant” of Gaussian Shading [1]. The frameworks of the two are very similar. The only obvious difference is that the DPM-solver part of Gaussian Shading is replaced with the sampler in the super-resolution model in this paper. As for this super-resolution model, its necessity is not seen. Both the image provided by the user and the watermarked images returned to the user are of normal resolution, which means that the so-called super-resolution model can be replaced by another stable-diffusion based model. We believe that both train-free and performance are advantages brought by stable diffusion, not the contribution of this paper.
- Performance is not good enough since visual quality and decoding accuracy is lower than some previous methods. According to Table 1, the PSNR of the proposed method is lower than CIN [2], MBRS [3] and so on, and the extraction accuracy is lower than StegaStamp [4] in most of the cases.
- As stated by the author, this paper also adopts the injection strategy of Gaussian Shade. We may raise a concern that, in the injection strategy of Gaussian Shade, there is one step that needs to replicate the watermarking information for many times, which may make the distribution deviate far from the uniform distribution. Therefore, an encryption process needs to be added to preserve a uniform distribution. Then a problem arises. When extracting the watermark, both the image and the key must be provided. Obviously, if there is a malicious attacker, we should assume that the attacker has obtained the same image. At this time, the security of the copyright is entirely entrusted to the key, so how can we ensure the security of the key? The service provider may also use a secure channel to transmit the key to the client at this time. This behavior is more like steganography than watermarking. After all, for watermarking, it is enough to extract the content that matches the identity. For previous methods, even if the attacker obtains the same image (from public channel), it is impossible to extract the content that matches him. But now the security guarantee has been transferred to the key, which needs to be transmitted over a secure channel. In this case, why not use a password, which is more convenient? We suggest the author include a section discussing the trade-offs between their approach and traditional watermarking in terms of security.

Ref:
- [1] Yang, Zijin, et al. "Gaussian Shading: Provable Performance-Lossless Image Watermarking for Diffusion Models." Proceedings of the IEEE/CVF Conference on Computer Vision and Pattern Recognition. 2024.
- [2] Ma, Rui, et al. "Towards blind watermarking: Combining invertible and non-invertible mechanisms." Proceedings of the 30th ACM International Conference on Multimedia. 2022.
- [3] Jia, Zhaoyang, Han Fang, and Weiming Zhang. "Mbrs: Enhancing robustness of dnn-based watermarking by mini-batch of real and simulated jpeg compression." Proceedings of the 29th ACM international conference on multimedia. 2021.
- [4] Tancik, Matthew, Ben Mildenhall, and Ren Ng. "Stegastamp: Invisible hyperlinks in physical photographs." Proceedings of the IEEE/CVF conference on computer vision and pattern recognition. 2020.

**Questions:**

See weaknesses.

---

### Official Review · Reviewer_AJhL · 2024-11-03

**Soundness:** 3
**Presentation:** 2
**Contribution:** 2
**Rating:** 5
**Confidence:** 4

**Summary:**

The paper presents SuperMark, a novel watermarking framework designed to address the challenges of robustness and fidelity in watermarking AI-generated and authentic content. It leverages insights from diffusion models to create a training-free approach that embeds watermarks into Gaussian noise and subsequently denoises this noise using pretrained super-resolution (SR) models. SuperMark demonstrates strong resilience to various distortions and adaptive attacks while maintaining high visual fidelity.

**Strengths:**

1. Flexibility and Transferability: SuperMark is adaptable across different datasets, SR models, watermark injection methods, and image resolutions, showcasing its versatility in practical applications.

2. No Fine-tuning Required: The ability to operate without fine-tuning the SR model simplifies the implementation and broadens accessibility for various use cases.

**Weaknesses:**

1. **Lack of innovation**: Key components of the proposed method, such as watermark injection techniques and the use of diffusion models, are not original. This limits the overall innovation of the research.
2. **Insufficient Contributions**: The claim that the study uncovers a critical insight into the encoder-noise layer-decoder architecture and its trade-offs lacks depth. The acknowledged trade-off between robustness and fidelity is already well-established; the authors do not offer a new perspective or solution to address this issue.
3. **Inadequate Experimental Comparisons**: The comparisons presented in Table 1 are neither significant nor comprehensive. For instance, the average accuracy on Normal Distortions does not show a significant lead over SepMark, and the performance on Adaptive Attacks is not the best. Furthermore, the comparison is unfair as it does not adequately control for visual quality (fidelity) when assessing robustness (accuracy). A more equitable comparison should maintain visual quality at similar levels.

**Questions:**

See weaknesses.

---

### Official Review · Reviewer_SsJy · 2024-11-04

**Soundness:** 2
**Presentation:** 3
**Contribution:** 2
**Rating:** 5
**Confidence:** 3

**Summary:**

This paper proposes a new watermarking framework with the consideration of both robustness and fidelity. The propose method includes a watermark embedding stage where the watermark is injected into the latent Gaussian noise and then a denoised procedure is followed to obtain the watermarked image; and a watermark extraction stage which involves a distortion process and a DDIM inversion to reconstruct the "watermarked noise", and further extract the watermark information. During these procedures, the models are frozen and no training/update is needed. The authors validate their method on different datasets as well as with different distortions.

**Strengths:**

There are several strengths as I can see.
1) The proposed method does not need to train (even fine-tune) models while it could work relatively well different datasets.
2) The method sounds simple but it works well. The authors also conduct a comprehensive ablation studies on important hyperparameters.
3) The writing and presentation are good. It is easy to follow.

**Weaknesses:**

There are several weaknesses/limitations of this version.
1) The novelty of this version is limited. Using diffusion models to do watermark is not new while this current version does not show sufficient empirical/theoretical evidence to distinguish itself from other works.
2) The experiments results shows that the proposed method is not the best performer in both sides---robustness and fidelity, thus the main contribution of this paper is limited.
3) Which pretrained VAE is used? It look like the authors do not mentioned it in the main paper.

**Questions:**

I have listed my concerns in the "weaknesses" section.

---

### Official Review · Reviewer_Wz7k · 2024-11-06

**Soundness:** 2
**Presentation:** 2
**Contribution:** 2
**Rating:** 5
**Confidence:** 3

**Summary:**

The paper proposed a new pipeline that uses stable diffusion to inject the watermark into the image. The main contribution of this paper is that it claims that it does not need training, and it is robust due to the characteristic of the stable diffusion model. It shows results comparable to those of the SOTA method and can prevent the attack from the VAE or diffusion model.

**Strengths:**

The method is innovative compared to the traditional encoder-decoder-based method. It leveraged the existing generative AI model to inject the watermark into the images, which reduced the additional cost of training a new model for the watermark. And it demonstrates reasonable performance.

**Weaknesses:**

The model does not compare the computation cost of the stable diffusion model with the existing methods. Stable diffusion is well-known for its intensive computation cost, slow inference speed, and large model size. If the stable diffusion model uses 10 times more MACs (Multiply-Accumulate Operations), FLOPs (Floating Point Operations), and large model size, while it performs similarly to traditional models, it diminishes the contribution of this method a lot. The model of adding a watermark is usually small and fast since it happens on the user's end. Most injections are conducted on mobile devices such as phones and laptops. The stable diffusion model will make it impossible to use in most scenarios. Overall, this is a good paper, but the completeness of the paper is not good enough. A lot of extensions mentioned in the paper could be done. This is more like an exploratory.

**Questions:**

1. Please compare the model size and computation cost with other methods for a fair comparison. Please separate the computation of adding and extracting watermarks for each method.
2. The PSNR and SSIM of this method are relatively low compared to other methods. However, stable diffusion should maintain high fidelity while maintaining a high watermark detection rate. This is why we want to use extra cost for stable diffusion, such as a large model for computing. If you train the diffusion model or fine-tune yourself, especially for the watermark task, stable diffusion should be able to maintain high fidelity.
3. Why not fine-tune a small diffusion model for injecting the watermark that can be used in mobile devices and another diffusion model for extracting the watermark that can be run on a large server? It makes the paper more complete and shows the superiority of stable diffusion. Since the method uses the most advanced techniques with large computation, it should outperform all the previous methods.

---

### Official Review · Reviewer_DCYr · 2024-11-06

**Soundness:** 1
**Presentation:** 2
**Contribution:** 1
**Rating:** 3
**Confidence:** 4

**Summary:**

A watermarking method is proposed for embedding watermarks in the diffusion generated images post-processing.  The proposed SuperMark embeds the watermark into initial Gaussian noise using existing techniques and then applies pre-trained Super-Resolution (SR) models to denoise the watermarked noise.

**Strengths:**

* The manuscript provides tests with specific "Adaptive Attacks".
* The supplementary material also provide numerous examples of watermarked images.

**Weaknesses:**

However, post-processing watermarking methods in general have been studied for long time since 1993 and the author only mentioned one method (Rahman, 2013) from the older times. They should have specified that such watermarking methods are spacial-based and frequency-based methods.

The claim of the limitation for other methods from the following sentence does not make sense and denotes that the authors do not have an overall understanding of the digital watermarking process.
"We reveal that their limitations are mostly stemmed from the disentanglement between robustness and fidelity due to the the encoder–noise layer–decoder architecture"
Actually, watermarked image quality, robustness, embedded information capacity and watermark security represent all requirements for a watermarking system and by enforcing each of these requirements actually limits the effects of all the others.

There is no support for the following claim, and the authors should provide theoretical or empirical proofs to support this:
"diffusion process holds inherent robustness against different distortions."

Also the authors do not provide enough discussion about their other claim that:
"diffusion process holds inherent robustness against different distortions."

The contributions claimed at the end Section 1 Introduction sound very general and do not actually represent major advancements in the areas in the way they are formulated.

Watermark embedding and watermark extraction stages from Sections 3.3 and 3.4 seem rather straightforward and do not bring any novelty. Sentences from Section 3.1 like "The Gaussian noise added to the conditioned image corresponds to the watermark information, allowing the watermark to be injected seamlessly" or "From this reconstructed noise, the watermark can be extracted effectively," are very general and meaningless. There is no new insight in the watermarking embedding and detection process.

For JPEG, additive noise and smoothing the results would have been better presented in some plots where various degrees of distortion caused by the attacks are tested.

The section about "Adoption of Tree-Ring’s watermark injection method" which adopts the method employed in Tree-Ring (Wen et al., 2024) is confusing. The method proposed in the Tree-Ring (Wen et al., 2024) is completely different from the one presented in this manuscript being embedded in the Fourier frequency domain, while this manuscript discusses a spatial embedding watermarking. The description from the Supplementary of Appendix A.1.2 Tree-ring is not that clear either, it mentions "low-frequency modes" but the description is very superficial and it is not clear how the frequencies are calculated.

**Questions:**

What is meant by "normal distortions" at the end of Section 1 Introduction? How much JPEG compression? What level of Gaussian noise?

The experimental results reported in Tables 1 or 3 are not clear, because they do not report the specific attacks used in the attacks. What does it mean JPEG? How much compression or quality factor was assumed? What level of "G Noise" or "G Blur"? What kind of "Crop" was applied?

It is claimed that
"We adopt an off-the-shelf strategy for watermark embedding, as used in Gaussian Shading (Yang et al., 2024),"
and this significantly limits the novelty of the proposed watermarking approach. What is actually the novelty?
The application on super-resolution images is not really a major advancement.

What is meant by "Tree-Ring is 0-bit watermark method"? First of all it is a Fourier domain watermarking method.

---

### Meta-Review · Area_Chair_BSAd · 2024-12-19

**Metareview:**

This paper works on watermarking. Authors proposed a robust and training-free watermarking framework. In the proposed method SuperMark, authors embed the watermark into initial Gaussian noise and then apply pretrained super-resolution models to denoise the watermarked noise to generate the final watermarked images. To extract the message, the watermarked image is converted back to the initial watermarked noise via DDIM Inversion.

8 reviewers unanimously rated this paper below the acceptance threshold.

Reviewers thought the strengths of this paper are: reduced additional cost for training.

Weaknesses of this paper are: some of the claims in the paper are not correct or supported; experimental results are limited or not enough; novelty or contribution is limited; performance is not good enough.

Authors didn't provide any rebuttal. so there is no discussion during the rebuttal period. Given these, AC decide to reject this paper.

**Additional Comments On Reviewer Discussion:**

Authors didn't provide any rebuttal.

---

### Decision · Program_Chairs · 2025-01-22

Reject